# The non-benzodiazepine anxiolytic etifoxine limits mechanical allodynia and anxiety-like symptoms in a mouse model of streptozotocin-induced diabetic neuropathy

**Géraldine Gazzo[ID], Marlene Salgado Ferrer, Pierrick Poisbeau[ID]***

Centre National de la Recherche Scientifique, Institute for Cellular and Integrative Neuroscience (INCI), University of Strasbourg, Strasbourg, France

\* poisbeau@unistra.fr

## Abstract

More than 450 million people worldwide suffer from diabetes, or 1 in 11 people. Chronic hyperglycemia degrades patients' quality of life and the development of neuropathic pain contributes to the burden of this disease. In this study, we used the mouse model of strepto-zocin-induced diabetic type 1 neuropathy to assess the analgesic potential of etifoxine. Eti-foxine is a prescribed anxiolytic that increases GABAAA receptor function through a direct positive allosteric modulation effect and, indirectly, by stimulating the production of endoge-nous GABAA receptor positive modulators such as allopregnanolone-type neurosteroids. We show that a post-symptomatic or preventive treatment strongly and durably reduces mechanical hyperalgesia and anxiety in diabetic neuropathic mice. This analgesic and neu-roprotective effect on painful symptoms and emotional comorbidities is promising and should now be clinically evaluated.

## Introduction

With 463 million affected people worldwide, prevalence of diabetes has drastically increased in the past few decades [1]. Painful diabetic neuropathy (PDN) is a frequent complication result-ing from diabetes, reaching an estimated 6–34% of patients and considerably contributing to the overall burden of this condition [2]. Clinical manifestations of PDN include painful symp-toms in the limbs (hyperalgesia or allodynia) often associated with unpleasant sensations such as paresthesia or numbness. On top of these somatic symptoms, PDN is also associated with an increased risk for the development of anxiety disorders [3,4], which dramatically worsen the patient's quality of life.

Although it is well accepted that prolonged hyperglycemia is the first step leading to nerve fiber damage, the detailed pathogenesis stages of diabetic neuropathy are still far from clear. Current leading hypotheses include the role of metabolic dysregulation leading to an increased activation of the polyol pathway and reactive oxygen species production contributing to oxi-dative stress [5]. Endothelial dysfunction, advanced glycation end-product deposition,

**Data Availability Statement:** As there are no ethical or legal restrictions on sharing, all data are freely available in the seafile depository server of

the University of Strasbourg as recommended by our research organization for open access. These data are now also accessible to the public under the following identifiers: - https://osf.io/9gp68/ - Poisbeau, Pierrick (2021): Gazzo et al_Plos_Fig2AC_openfield-light-marble.pzfx. figshare. Dataset. https://doi.org/10.6084/m9.figshare.15022608.v2 - Poisbeau, Pierrick (2021): Gazzo et al_Plos_Fig1AB_preventif-curatif.pzfx. figshare. Dataset. https://doi.org/10.6084/m9.figshare.15022605.v1.

**Funding:** The following institutions gave their support to the project: Centre National de la Recherche Scientifique and Université de Strasbourg. This work was funded by the French National Research Agency (ANR) through the Programme d'Investissement d'Avenir under the contract ANR-17-EURE-0022. The present study was specifically conducted under the frame of a CNRS collaborative research contract with Biocodex laboratories.

**Competing interests:** PP received financial support from Biocodex laboratories to investigate the molecular mechanisms of action of etifoxine. The financial support by Biocodex laboratories does not alter adherence to PLOS ONE policies on sharing data and materials. In good agreement, data could be filed in an open source depository. Moreover, the funders had no role in study design, data collection and analysis, decision to publish, or preparation of the manuscript.

pro-inflammatory processes and neurotrophic factor deficiency also are considered as major contributing factors [6,7]. Although multifactorial causes conjointly lead to PDN, it is interesting to note here that hyperglycemia-induced mitochondrial dysfunction appears as a key player in PDN development, contributing to the production of free radicals, activation of cell death pathways, and responsible for a depletion in ATP synthesis [7].

In this context, we explored the therapeutic potentiel of etifoxine (EFX) in the treatment of PDN pain symptoms and comorbid anxiety. EFX is a non-benzodiazepine anxiolytic prescribed in several countries for the treatment of adaptation disorders with anxiety [8–10]. On top of acting as a positive allosteric modulator of $GABA_A$ receptors [11], EFX also binds to the mitochondrial 18-kDa translocator protein (TSPO) complex, favoring cholesterol entry in the mitochondria and subsequent neurosteroid production [12–14]. This action of EFX on neurosteroidogenesis has been shown to limit pain symptoms in several preclinical models [15]. Indeed, EFX has shown analgesic properties in animals models of neuropathic pain [16,17], and also prevented the apparition of anxiodepressive-like comorbidities in a model of mononeuropathy following constriction of the sciatic nerve [18]. Furthermore, EFX also presents neuroprotective actions and promotes nerve regeneration in several rodent models [19–22], suggesting a potential therapeutic interest in the treatment of PDN.

At the present date, prevention and limitation of diabetic neuropathy evolution mainly relies on glycemia control [23,24], while pain management largely depends on the use of large spectrum antalgic drugs, anticonvulsants or antidepressants [5,6], without providing full patient satisfaction. Considering the aforementioned properties of EFX, the aim of this project thus was to evaluate the properties of post-symptomatic or preventive etifoxine in the relief of PDN pain and anxiety-like symptoms in a rodent model of type 1 diabetes-induced neuropathy.

## Materials & methods

### Animals

Experiments were performed on adult male C57BL6J mice (Charles River, France) aged 8–12 weeks at the time of neuropathic pain induction. Animals were housed in a temperature (23 ± 1°C) and humidity (50 ± 10%) controlled room under a 12h light-dark cycle (lights on at 7:00am). Animals were housed in group cages with *ad libitum* access to food and tap water. All procedures were conducted in accordance with EU regulations and approved by the local ethical committee (CREMEAS, Comité Régional d'Ethique en Matière d'Expérimentation Animale de Strasbourg: authorization number 2016110716292742). At the end of all experimental procedures, animals were sacrificed with cervical dislocation, performed by appropriately trained and competent personnel.

### Streptozotocin-induced diabetic neuropathy

Type 1 diabetes was induced with a single intraperitoneal (i.p.) injection of 150 mg/kg streptozotocin (STZ; Merck, France) freshly dissolved in 0,9% NaCl, at a volume of 0,1 mL/10 g. CTRL animals received a single i.p. injection of 0,9% NaCl, vehicle for STZ [25]. A week after STZ injection, hyperglycemia was evaluated using a glucometer (Accu-Chek Performa, Accu-Check, France) with 5 μL blood samples collected from one of the lateral caudal veins. Only animals which presented non fasting blood glucose levels ≥ 2,25 g/L were considered diabetic and kept in the STZ group.

## Pharmacological treatment

Etifoxine (EFX; 2-ethylamino-6-chloro-4-methyl-4-phenyl-4H-3,1-benzoxazine hydrochloride) was kindly provided by Biocodex laboratories (batch 653; Biocodex, Gentilly, France). EFX was prepared in 0,9% NaCl containing 1,5% ethanol and 1% Tween 80 (Merck, France) and administered i.p. (0,1 mL/kg) at a dose of 50 mg/kg [17]. Control animals received an equivalent volume of vehicle. STZ and VEH animals were randomly assigned to the EFX or VEH group, after verification that the difference in mechanical nociceptive baseline was not significant in-between both conditions in the established treatment groups. Experimenters were then blind to the treatment condition when performing behavioral assays.

## Evaluation of mechanical nociceptive sensitivity

Von Frey filaments were used (Stoelting, Wood Dale, IL, USA) according to a protocol adapted from Chaplan [26]. Animals were placed in clear Plexiglas® boxes (7 × 9 × 7 cm) on an elevated mesh screen. After 15 min habituation, calibrated von Frey filaments were applied on the plantar surface of each hindpaw in a series of ascending forces. Each filament was tested 5 times per paw, and the mechanical nociceptive threshold was considered to correspond to the force of the first von Frey filament eliciting 3 or more withdrawals of the paw out of the five trials [27].

## Evaluation of anxiety-like symptoms

**Light/Dark box.** Apparatus consisted in two compartments (20 x 20 x 15 cm each), one dark and one brightly lit (350 lux), connected by a dark tunnel (7 x 7 x 10 cm). Animals were placed in the lit compartment and video-tracked for 5 min. Time spent in the lit compartment was analyzed with the ANY-maze 5.2 software (Ugo Basile, Gemonio, Italy).

**Open field.** Animals were placed nose facing one of the walls of an open field (40 x 40 x 30 cm) lit at 120 lux. Animals were video-tracked for 5 minutes, during which time spent in the center of the open field (24 x24 cm) and total distance travelled in the whole apparatus were analyzed with the ANY-maze 5.2 software (Ugo Basile, Gemonio, Italy).

**Marble burying test.** Mice were placed individually in Plexiglas® cages (27 x 16 x 14 cm) containing 3 cm of sawdust on top of which twenty-five glass marbles (diameter 1 cm) were evenly placed. After being left undisturbed for 30 min, animals were removed from the cage and the number of buried marbles were counted by an observer blind to the condition of the animals. Marbles were considered buried if two thirds or more of their surface was covered by sawdust. The number of buried marbles is considered a measure of animal anxiety and also reflects obsessive compulsive disorders [28].

## Statistical analysis

Data are expressed as mean ± standard error of the mean (SEM). Statistical analysis was performed using the GraphPad Prism 6 software (Lajolla, CA, USA). Normal distribution of values was verified with the Shapiro-Wilk normality test before performing parametric analysis. Two-way (time x condition) analysis of variance tests (ANOVA), with repeated measures for the time variable (2wRM-ANOVA) were used to evaluate the time course of pain thresholds and followed by Tukey's post hoc multiple comparison test. Anxiety-like symptoms were assessed with an ordinary 2w-ANOVA (condition x treatment) followed by Tukey's post-hoc test. Differences were considered to be statistically significant for $p < 0.05$.

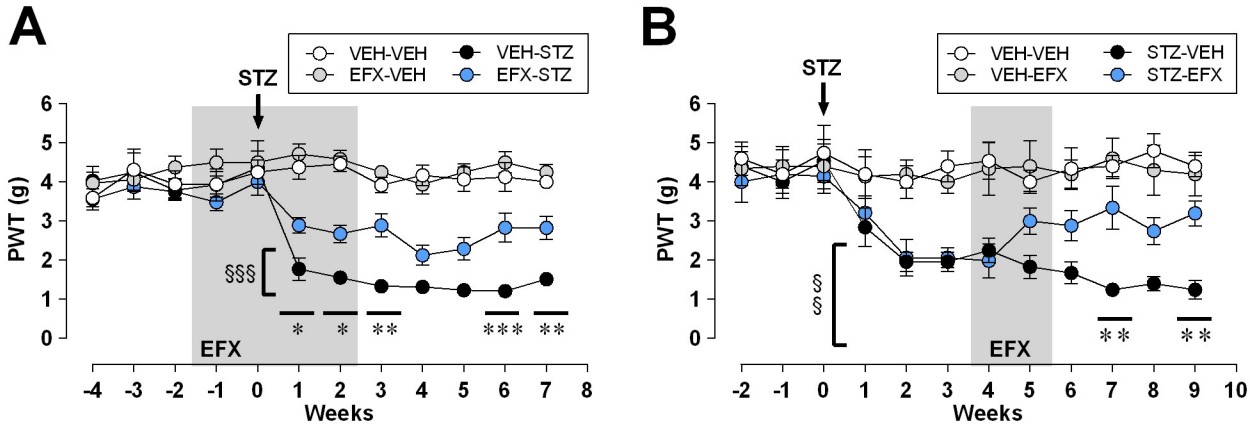

**Fig 1. Preventive (A) and post-symptomatic (B) etifoxine action on von Frey paw withdrawal thresholds (PWT) following STZ-induced diabetic neuropathy (n = 10 per group).** STZ injection was performed at week 0 (arrows), and treatment period is indicated in grey for a 4-week preventive (panel A) and a 2-week post-symptomatic EFX treatment (panel B). Statistical significance was assessed with Tukey's multiple comparison test, illustrated as follows: $p < 0.05$ (*), $p < 0.01$ (**) or $p < 0.001$ (***) for comparisons between VEH and EFX-treated STZ groups and $p < 0.01$ (§§), $p < 0.001$ (§§§) for comparisons between STZ-VEH and its control VEH-VEH at each time point.

## Results

### EFX as a preventive compound limiting the extent of STZ-induced mechanical allodynia

Evolution of mechanical thresholds was assessed in a model of STZ-induced diabetic neuropathy (Fig 1A). STZ injection induced a progressive decrease in mechanical thresholds that became significantly lower than in the control groups after one week, with PWT dropping to $1.77 \pm 0.29$ g, compared to $4.38 \pm 0.30$ g in control animals (2wRM-ANOVA (time x condition), $F_{(33,638)} = 7.450$, $p < 0.0001$). Mechanical allodynia persisted until the end of the observation period, i.e. 7 weeks after STZ injection, with a 62% decrease in PWT compared to VEH-VEH animals at the same time point.

A first group of animals received a preventive EFX treatment, which started two weeks before STZ injection and was continued until the end of the second week following STZ injection. EFX significantly reduced STZ-induced mechanical allodynia, with PWT that were significantly higher in EFX-treated compared to non-treated STZ-animals, as soon as the first week following STZ injection ($2.89 \pm 0.19$ g in EFX-STZ vs $1.77 \pm 0.29$ g in VEH-STZ at week 1). This analgesic effect persisted long after cessation of treatment since PWT were of $2.83 \pm 0.3$ g in EFX-STZ animals five weeks after the end of the treatment (i.e. week 7), significantly higher than the $1.51 \pm 0.14$ g threshold displayed by VEH-STZ animals. Although EFX successfully limited mechanical pain symptoms, it did not completely prevent PDN development since PWT remained lower in EFX-treated STZ animals compared to VEH-VEH and EFX-VEH control groups.

### Durable analgesic effect of EFX on STZ-induced mechanical allodynia

In another group of animals, EFX treatment was started 4 weeks after STZ injection in order to allow for pain symptoms to develop (Fig 1B). Animals were then exposed to two sessions of 5 consecutive daily injections separated by two days, hence treatment lasted for a period of two weeks. EFX treatment durably increased von Frey thresholds in the STZ group, with values that were significantly higher than in VEH-treated STZ animals on the 7th and 9th weeks following STZ injection (2wRM-ANOVA (time x condition), $F_{(33,396)} = 2.196$, $p = 0.0002$). On

the 9th week following STZ injection for example, PWT were 3.2 ± 0.33 g in STZ-EFX animals, significantly higher than the 1.24 ± 0.24 g threshold displayed by STZ-VEH animals.

### Effect of EFX on anxiety-like symptoms

We then evaluated anxiety-like symptoms following PDN apparition (Fig 2). In the open field test, performed 6 weeks after STZ injection, neuropathic animals displayed a moderate tendency towards a decrease in the time spent in the anxiogenic center of the open field (Fig 2A₁; 2w-ANOVA (condition x treatment), STZ condition factor: $F_{(1,42)} = 1.088$, p = 0.3028). EFX-treated STZ animals spent 17.64 ± 5.29 s in the center of the OF, while STZ-VEH only spent 11.1 ± 3.01 s, which could indicate a non-significant tendency towards a decrease in anxiety-like signs following EFX treatment (2w-ANOVA (condition x treatment), EFX treatment factor: $F_{(1,42)} = 1.056$, p = 0.3099). Total distance travelled in the open field during the 5-minute test show no significant difference between groups (Fig 2A₂; 2w-ANOVA (condition x treatment), $F_{(1,39)} = 0.0003967$, p = 0.9842).

In the light/dark box test however, we were unable to demonstrate any anxiety-like signs 7 weeks after STZ injection, since no difference was found in the time spent in the light chamber between VEH-VEH and STZ-VEH animals (Fig 2B; 2w-ANOVA (condition x treatment), STZ condition factor: $F_{(1,58)} = 2.789$, p = 0.1003). The anxiolytic effect of EFX was however significant in both STZ and VEH groups (2w-ANOVA (condition x treatment), EFX treatment factor: $F_{(1,58)} = 14.90$, p = 0.0003). Indeed, EFX significantly increased the time spent in the light chamber, both in neuropathic (46.36 ± 6.61 s in STZ-VEH vs 76.96 ± 7.83 s in STZ-EFX) and control animals (59.98 ± 4.78 s in VEH-VEH vs 89.25 ± 11.74 s in VEH-EFX).

Finally, as illustrated in Fig 2C, non-treated animals with STZ-induced PDN buried a mean of 2.71 ± 1.12 marbles in the 30-min session, significantly less than the 14.06 ± 1.34 marbles buried by VEH-VEH animals (2w-ANOVA (condition x treatment), $F_{(1,26)} = 8.114$, p = 0.0085). Here, EFX treatment significantly restored the number of buried marbles in EFX-treated STZ animals to a mean of 10.92 ± 3.31 marbles, similar to the mean number buried in the control groups.

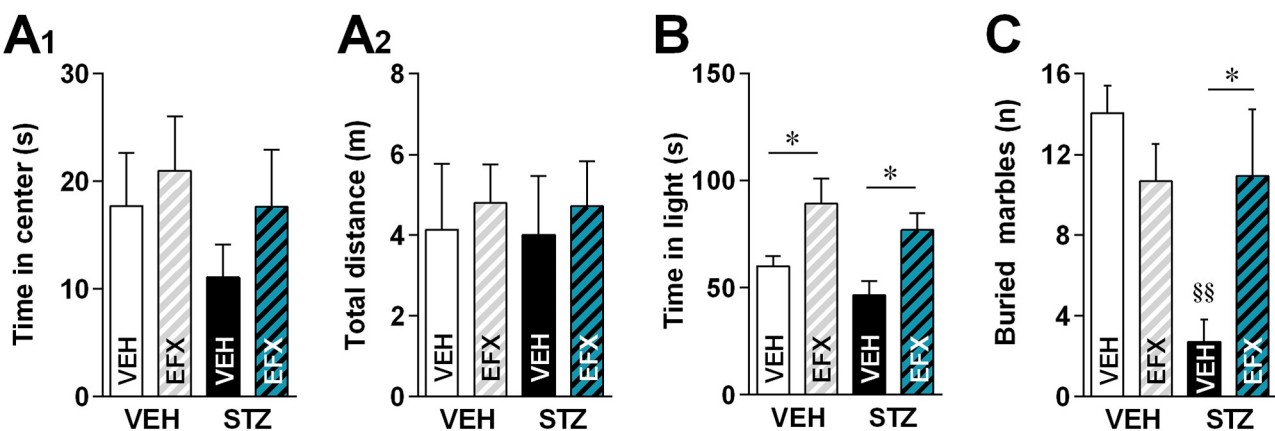

**Fig 2. Effect of etifoxine on anxiety-like symptoms following STZ-induced diabetic neuropathy.** A. Time spent in the center of an open field (A₁; n = 10–14 per group) and total distance travelled in the open field (A₂). B. Time spent in the light compartment of a light/dark box (n = 13–21 per group). C. Mean number of buried marbles (n = 7–8 per group). Statistical significance was assessed with Tukey's multiple comparison test, illustrated as follows: p < 0.05 (*) for intra-group comparisons (VEH vs EFX treatment) and p < 0.01 (§§) for inter-group comparisons (VEH or EFX).

## Discussion

Here, we showed that EFX, administered after PDN development, or preventively prior to neuropathy induction, successfully and durably limited STZ-induced mechanical allodynia. EFX also showed a tendency towards an anxiolytic effect in STZ animals, although we were not able to observe strong anxiety-like comorbidities in this model of PDN.

The anti-allodynic effect of EFX observed in this model is in accordance with other studies in which EFX successfully limited pain symptoms [15]. Long-lasting analgesic effect was previously shown to rely on the promotion of neurosteroidogenesis following EFX binding to the mitochondrial TSPO complex. Considering the alteration of mitochondrial function induced by diabetes mellitus [29,30], restoring mitochondrial neurosteroidogenesis could be an interesting strategy in order to prevent long-term damage induced by a poor mitochondrial function [31–33].

Microglial activation in the spinal cord has been reported in rat models of STZ-induced type 1 diabetes [34,35], which could contribute to sensory disturbances and an increased production of pro-inflammatory cytokines [6]. In this context, EFX has shown beneficial effects in the reduction of inflammatory pain symptoms in models of knee monoarthritis [36] or carrageenan-induced inflammatory sensitization [37]. Therefore, the analgesic properties of EFX observed here in a model of STZ-induced PDN could be due, in part, to a reduction of pro-inflammatory cytokine production and microglial activation [17]. Similarly, preventive EFX could have limited nerve damage considering EFX's demonstrated neuroprotective properties [20,22,38].

Diabetic neuropathy patients often exhibit anxiodepressive comorbidities, as it is the case in many chronic pain states [3,39]. In this study, we were not fully successful to demonstrate any strong anxiety-like signs in our model using the open field and light/dark box tests. However, sharp differences could be seen in the marble burying test which is also used to reveal stereotypic behavior abnormalities often associated with obsessive-compulsive disorders [40]. Usually, a high number of marbles indicates a strong anxiety-like phenotype [28]. In our study, results are hard to interpret as STZ animals buried a very low amount of marbles, contrary to what was expected, while the anxiolytic EFX restored control values. Nonetheless, this result suggests that EFX benefited STZ animals since it brought back values closer to that of control animals.

Altogether, we provide further evidence that EFX could be a useful drug to alleviate pain symptoms and emotional comorbidities in this model of PDN. Due to the mechanism of action of EFX, further investigations will be required to ensure its use since neuropathic states resulting from metabolic dysfunction may alter the efficacy of drugs such as EFX. It remains that EFX was efficient to alleviate pain responses and anxiogenic behaviors in this model. Clinical trials using this already prescribed anxiolytic will help confirm the therapeutic potential of EFX.

## Acknowledgments

We thank Stéphane Doridot, Edouard Gottschalk and all other members of the Chronobiotron facility for the animal care provided. GG and MSF were enrolled as fellows of the graduate school of pain EURIDOL (ANR-17-EURE-0022). We thank Biocodex laboratories (Gentilly, France) for kindly providing us with etifoxine. GG and MSF were both PhD fellows enrolled in the.

## Author Contributions

**Conceptualization:** Géraldine Gazzo, Pierrick Poisbeau.

**Data curation:** Marlene Salgado Ferrer.

**Formal analysis:** Géraldine Gazzo, Marlene Salgado Ferrer, Pierrick Poisbeau.

**Funding acquisition:** Pierrick Poisbeau.

**Investigation:** Géraldine Gazzo, Marlene Salgado Ferrer, Pierrick Poisbeau.

**Methodology:** Géraldine Gazzo, Pierrick Poisbeau.

**Project administration:** Pierrick Poisbeau.

**Supervision:** Pierrick Poisbeau.

**Validation:** Pierrick Poisbeau.

**Writing – original draft:** Géraldine Gazzo, Pierrick Poisbeau.

**Writing – review & editing:** Géraldine Gazzo, Pierrick Poisbeau.

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
