## [Decision Letter · Decision Letter 0]

26 May 2021

PONE-D-21-05383

The non-benzodiazepine anxiolytic etifoxine limits the sensory-affective expression of pain following streptozotocin-induced diabetic neuropathy

PLOS ONE

Dear Dr. 

Thank you for submitting your manuscript to PLOS ONE. After careful consideration, we feel that it has merit but does not fully meet PLOS ONE’s publication criteria as it currently stands. Therefore, we invite you to submit a revised version of the manuscript that addresses the points raised during the review process.

We look forward to receiving your revised manuscript.

Kind regards,

Rosanna Di Paola, MD

Academic Editor

PLOS ONE

Journal Requirements:

2. To comply with PLOS ONE submissions requirements, please provide methods of sacrifice in the Methods section of your manuscript.

4. Thank you for stating the following in the Financial Disclosure section:

'The following institutions gave their support to the project: Centre National de la Recherche Scientifique and Université de Strasbourg. This work was funded by the French National Research Agency (ANR) through the Programme d’Investissement d’Avenir under the contract ANR-17-EURE-0022. The present study was specifically conducted under the frame of a CNRS collaborative research contract with Biocodex laboratories. '

We note that you received funding from a commercial source: Biocodex laboratories.

Within this Competing Interests Statement, please confirm that this does not alter your adherence to all PLOS ONE policies on sharing data and materials by including the following statement: "This does not alter our adherence to PLOS ONE policies on sharing data and materials.” (as detailed online in our guide for authors http://journals.plos.org/plosone/s/competing-interests).  If there are restrictions on sharing of data and/or materials, please state these.

Please note that we cannot proceed with consideration of your article until this information has been declared.

c. Please include your amended Competing Interests Statement and Role of Funder statement within your cover letter. We will change the online submission form on your behalf.

Reviewers' comments:

Reviewer's Responses to Questions

**Comments to the Author**

1. Is the manuscript technically sound, and do the data support the conclusions?

Reviewer #1: Yes

Reviewer #2: Yes

2. Has the statistical analysis been performed appropriately and rigorously? 

Reviewer #1: Yes

Reviewer #2: Yes

3. Have the authors made all data underlying the findings in their manuscript fully available?

Reviewer #1: Yes

Reviewer #2: Yes

4. Is the manuscript presented in an intelligible fashion and written in standard English?

Reviewer #1: Yes

Reviewer #2: Yes

5. Review Comments to the Author

Reviewer #1: The manuscript investigates the effects of the non-benzodiazepine anxiolytic Etifoxine on mechanical allodynia and anxiety in a mouse model of painful diabetic neuropathy induced by streptozotocin (STZ). The Authors demonstrates that Etifoxine prevents and reverses mechanical allodynia and anxiety in the STZ mice.

The results are intriguing and could lead to a novel and effective treatment for painful diabetic neuropathy, a prevalent and significant health issue. The paper is well written and the logic of the experiments is easy to follows.

Major points:

1) The Authors should describe whether the experimental set up included male and female mice and if the experimental design allowed for disaggregation of data so that results obtained from males and females can be analyzed separately and compared. These days it is not acceptable to just perform experiments on male animals. It has been shown that males and females have extremely different pain responses driven by different factors in some instances.

2) Blinding is not mentioned in the behavior studies. Given that blinding is critical for these experiments, authors should describe whether and how blinding was included in the experimental design. In addition, there is not mention of randomization methods in the experimental design. These factors weaken the ability of the Authors to reproduce the work.

3) The STZ model is a model of type 1 diabetes. Most commonly, however, painful neuropathy is associated with type 2 diabetes. In order for the study to be more impactful, the Authors should confirm their findings in models of painful diabetic neuropathy associated with type 2 diabetes, such as the High-Fat-Diet mouse model or db-db obese mice.

Minor points

1) The Authors should rework the title. It is not clear what the Authors mean for “sensory-affective expression of pain”. The Authors in the study are measuring measure static tactile reflexes; a potential measure related to tactile allodynia in addition to anxiety.

2) The Authors could present the preventive experiment first and then conclude the paper with the reversal experiment. The reversal is even more relevant from a translational perspective.

3) The Authors should not use the term: “curative”. Indeed, the Authors are not providing evidence of disease modification, such as reversal of small fiber degeneration etc.

Reviewer #2: Interesting manuscript, even if the topic cannot be regarded as very original. The title is too generic. Please, specify in the title that the study has been carried out in rats or at least specificy streptozotocin-induced diabetic neuropathy "model". The Introduction is just a little bit long because the authors should draw readers' attention to simply general overview of PDN and few specific statements about EFX. Materials&methods is well-written and statistical analysis is accurate. Results, supported by statistics, are easily "visualized" owing to well-constructed figures and well-detailed legends. The Discussion is to shorten because too redundant. Authors should more focus on the effect of etifoxine against PDN, the topic of their study, and less wander off. I would suggest authors to give some more details about microglia activation and neuropathic chronic states due to metabolic dysfunction, trying to better clarify the role of EFX.

Reference list is good and updated.

6. PLOS authors have the option to publish the peer review history of their article (what does this mean?). If published, this will include your full peer review and any attached files.

Reviewer #1: No

Reviewer #2: No

---

## [Author Response · Author response to Decision Letter 0]

5 Jul 2021

Dear Editor of Plos one,

Please find below our detailed answers to the reviewers comments. We thank them for their constructive comments, which help us revise the manuscript. We do hope that our answers will clarify the concerns raised after their reading. Below are the point-by-point answers preceded by some editorial changes we were asked to do. 

Answer to Editor’s requests:

1. Style of the journal: We have checked all parts in order to respect the recommended style. 

2. Sacrifice method was added to the “animals” paragraph of the materials and methods section.

3. As there is non ethical or legal restrictions on sharing, all data are freely available in the seafile depository server of the University of Strasbourg as recommended by our research organization for open access. These data are now also accessible to the public under the following DOI number : 10.17605/OSF.IO/9GP68

4. We have transferred the funding bodies informations to the Financial Disclosure section. 

5. A competing interest statement is included in the submission. It indicates that the financial support by Biocodex laboratories does not alter adherence to PLOS ONE policies on sharing data and materials. This is why data could be filed in an open source depository. Moreover, the funders had no role in study design, data collection and analysis, decision to publish, or preparation of the manuscript.

Answer to Reviewers' comments:

Reviewer #1: The manuscript investigates the effects of the non-benzodiazepine anxiolytic Etifoxine on mechanical allodynia and anxiety in a mouse model of painful diabetic neuropathy induced by streptozotocin (STZ). The Authors demonstrates that Etifoxine prevents and reverses mechanical allodynia and anxiety in the STZ mice.

The results are intriguing and could lead to a novel and effective treatment for painful diabetic neuropathy, a prevalent and significant health issue. The paper is well written and the logic of the experiments is easy to follows.

Major points:

1) The Authors should describe whether the experimental set up included male and female mice and if the experimental design allowed for disaggregation of data so that results obtained from males and females can be analyzed separately and compared. These days it is not acceptable to just perform experiments on male animals. It has been shown that males and females have extremely different pain responses driven by different factors in some instances.

We completely agree with this crucial point and deplore the lack of studies performed on both male and female rodent models, especially considering the mentioned differential responses to pain and mechanisms of chronicization in males and females. The present study however only is a brief research report meant as a proof of concept of the potential effect of Etifoxine following STZ-induced diabetic neuropathy. This statement is in no case an excuse, however, considering the reported increased resistance of female rodents to STZ-induced diabetes (reviewed in Deeds et al. 2011 - https://www.ncbi.nlm.nih.gov/pmc/articles/PMC3917305/), which could be due in part to the protective action of estrogens (Le May et al., 2006 - https://pubmed.ncbi.nlm.nih.gov/16754860/), we decided to pursue our brief proof of concept in highly-susceptible male mice only, as a first step in a broader study. 

2) Blinding is not mentioned in the behavior studies. Given that blinding is critical for these experiments, authors should describe whether and how blinding was included in the experimental design. In addition, there is not mention of randomization methods in the experimental design. These factors weaken the ability of the Authors to reproduce the work.

Following the reviewer’s suggestion, we have added information in this regard in the “pharmacological treatment” section of the materials and methods. Blinding could not be included in the STZ vs VEH group considering the STZ-induced weight loss that even a blind experimenter could recognize. However, attribution of EFX or VEH treatment between CTRL and STZ groups was randomized. As added in the materials and methods, “STZ and VEH animals were randomly assigned to the EFX or VEH group, after verification that the difference in mechanical nociceptive baseline was not significant in-between both conditions in the established treatment groups. Experimenters were then blind to the treatment condition when performing behavioral assays”.

3) The STZ model is a model of type 1 diabetes. Most commonly, however, painful neuropathy is associated with type 2 diabetes. In order for the study to be more impactful, the Authors should confirm their findings in models of painful diabetic neuropathy associated with type 2 diabetes, such as the High-Fat-Diet mouse model or db-db obese mice. 

We agree with the reviewer, and had performed preliminary experiments in db/db-/- mice. However, our behavioral testing procedures were not calibrated for severely obese animals and we failed to highlight any significant results in this model.

Minor points

1) The Authors should rework the title. It is not clear what the Authors mean for “sensory-affective expression of pain”. The Authors in the study are measuring measure static tactile reflexes; a potential measure related to tactile allodynia in addition to anxiety.

The title has been reworked and specified and now states “The non-benzodiazepine anxiolytic etifoxine limits mechanical allodynia and anxiety-like symptoms in a mouse model of streptozotocin-induced diabetic neuropathy”.

2) The Authors could present the preventive experiment first and then conclude the paper with the reversal experiment. The reversal is even more relevant from a translational perspective.

We took this comment into account and presented the preventive experiment first. 

3) The Authors should use the term “curative”. Indeed, the Authors are not providing evidence of disease modification, such as reversal of small fiber degeneration etc.

We took this comment into account. Since we only meant “curative” as a timely adjective in opposition to “preventive”, we replaced the term “curative” with the term “post-symptomatic”.

Reviewer #2: Interesting manuscript, even if the topic cannot be regarded as very original. The title is too generic. Please, specify in the title that the study has been carried out in rats or at least specificy streptozotocin-induced diabetic neuropathy "model". The Introduction is just a little bit long because the authors should draw readers' attention to simply general overview of PDN and few specific statements about EFX. Materials&methods is well-written and statistical analysis is accurate. Results, supported by statistics, are easily "visualized" owing to well-constructed figures and well-detailed legends. The Discussion is to shorten because too redundant. Authors should more focus on the effect of etifoxine against PDN, the topic of their study, and less wander off. I would suggest authors to give some more details about microglia activation and neuropathic chronic states due to metabolic dysfunction, trying to better clarify the role of EFX. Reference list is good and updated.

The authors thank the reviewer for their comments. 

* Title has been modified to specify that the study was performed in a mouse model and now states “The non-benzodiazepine anxiolytic etifoxine limits mechanical allodynia and anxiety-like symptoms in a mouse model of streptozotocin-induced diabetic neuropathy”.

* The introduction was not shortened as we feel that an overview of EFX’s general mechanism of action relative to the pathogenesis of PDN allows the reader to understand why we sought to evaluate the properties of EFX in a model of STZ-induced neuropathy.

* The discussion was shortened following reviewer’s comment as we had no intention to appear as if we were wandering off.

---

## [Editor Report · Decision Letter 1]

15 Jul 2021

The non-benzodiazepine anxiolytic etifoxine limits mechanical allodynia and anxiety-like symptoms in a mouse model of streptozotocin-induced diabetic neuropathy

PONE-D-21-05383R1

Dear Dr.

We’re pleased to inform you that your manuscript has been judged scientifically suitable for publication and will be formally accepted for publication once it meets all outstanding technical requirements.

Kind regards,

Rosanna Di Paola, MD

Academic Editor

PLOS ONE
---

## [Editor Report · Acceptance letter]

27 Jul 2021

PONE-D-21-05383R1 

The non-benzodiazepine anxiolytic etifoxine limits mechanical allodynia and anxiety-like symptoms in a mouse model of streptozotocin-induced diabetic neuropathy 

Dear Dr. Poisbeau:

I'm pleased to inform you that your manuscript has been deemed suitable for publication in PLOS ONE. Congratulations! Your manuscript is now with our production department. 

Kind regards, 

on behalf of

Dr. Rosanna Di Paola 

Academic Editor

PLOS ONE